# Motor Cortical Correlates of Paired Associative Stimulation Induced Plasticity: A TMS-EEG Study

**DOI:** 10.3390/brainsci13060921

**Published:** 2023-06-07

**Authors:** Matteo Costanzo, Giorgio Leodori, Carolina Cutrona, Francesco Marchet, Maria Ilenia De Bartolo, Marco Mancuso, Daniele Belvisi, Antonella Conte, Alfredo Berardelli, Giovanni Fabbrini

**Affiliations:** 1IRCCS Neuromed, 86077 Pozzilli, Italy; matteo.costanzo@uniroma1.it (M.C.); giorgio.leodori@uniroma1.it (G.L.); carolina.cutrona@uniroma1.it (C.C.); mariailenia.debartolo@uniroma1.it (M.I.D.B.); daniele.belvisi@uniroma1.it (D.B.); antonella.conte@uniroma1.it (A.C.); alfredo.berardelli@uniroma1.it (A.B.); 2Department of Human Neurosciences, Sapienza University of Rome, Viale dell’Università 30, 00185 Rome, Italy; francesco.marchet@uniroma1.it (F.M.); marco.mancuso@uniroma1.it (M.M.)

**Keywords:** transcranial magnetic stimulation, paired associative stimulation, PAS, cortical plasticity, cortical correlates, TMS-EEG, TMS-evoked potentials, TEPs, sensorimotor integration

## Abstract

Paired associative stimulation (PAS) is a non-invasive brain stimulation technique that modulates synaptic plasticity in the human motor cortex (M1). Since previous studies have primarily used motor-evoked potentials (MEPs) as outcome measure, cortical correlates of PAS-induced plasticity remain unknown. Therefore, the aim of this observational study was to investigate cortical correlates of a standard PAS induced plasticity in the primary motor cortex by using a combined TMS-EEG approach in a cohort of eighteen healthy subjects. In addition to the expected long-lasting facilitatory modulation of MEPs amplitude, PAS intervention also induced a significant increase in transcranial magnetic stimulation-evoked potentials (TEPs) P30 and P60 amplitude. No significant correlation between the magnitude of PAS-induced changes in TEP components and MEP amplitude were observed. However, the linear regression analysis revealed that the combined changes in P30 and P60 component amplitudes significantly predicted the MEP facilitation after PAS. The findings of our study offer novel insight into the neurophysiological changes associated with PAS-induced plasticity at M1 cortical level and suggest a complex relationship between TEPs and MEPs changes following PAS.

## 1. Introduction

Paired associative stimulation (PAS) is a non-invasive brain stimulation technique known to modulate synaptic plasticity in the human motor cortex [1,2]. A standard PAS protocol consists of repetitive pairs of peripheral nerve electrical stimulation followed by transcranial magnetic stimulation (TMS) of the primary motor cortex (M1). The interstimulus interval is designed to generate near-synchronous input to M1 [2]. The PAS paradigm effectively induces a fast developing, enduring, and stimulus-specific enhancement in corticomotor excitability [2]. This facilitation has been suggested to operate through long-term synaptic potentiation-like mechanisms, with M1′s horizontal cortico-cortical connections as a possible neural substrate [2,3,4]. Beside cortical mechanisms, spinal mechanisms have also been suggested to play a role [5]. In healthy humans, PAS-induced plasticity has been suggested by showing an increase in the amplitude of the motor-evoked potential (MEP), as measured by electromyography (EMG) [2,3,4,5,6,7]. However, since MEPs reflect the excitability of a limited cortical circuit directly projecting on pyramidal tract neurons, as well as spinal mechanisms, it becomes challenging to determine the cortical mechanisms of PAS-induced plasticity based on MEPs only.

A deeper comprehension of the cortical mechanisms underpinning the PAS paradigm would enhance its application both as an investigative tool for cortical plasticity and as a potential treatment option. Recent developments have now made possible the recording of electroencephalographic (EEG) activity elicited by TMS as TMS-evoked potentials (TEPs). TEPs from M1 stimulation consist of a time-locked series of peaks and troughs lasting about 300 ms, reflecting M1 local excitability and its effective connectivity [8,9,10,11,12]. TMS-EEG has emerged as a non-invasive neurophysiological technique useful for investigating cortical correlates of TMS-based measures [9,13,14,15,16,17]. Studies employing paired-pulse TMS protocols to assess M1 inhibitory and excitatory intracortical circuits have revealed specific modulations in the early and late components of TEPs that were, in part, unrelated to MEP changes [15,16,18]. These observations highlight the value of TEPs as a tool to investigate both inhibitory and excitatory neural circuits at M1 level that might be missed when relying solely on MEPs [19,20,21,22,23]. Hence, TEPs serve as a promising tool for exploring the mechanisms of PAS-induced plasticity [13].

In this observational study, we aimed to investigate the cortical correlates of a standard PAS-induced plasticity protocol on M1 by using a combined TMS-EEG approach in healthy humans. To this end, we recorded MEPs and TEPs before and after a PAS intervention, as well as the TEPs evoked during the PAS protocol. No study has investigated the cortical correlates of PAS by means of TEPs. However, previous studies showed that short-latency afferent inhibition (SAI), a TMS paradigm resulting in MEP inhibition through sensorimotor integration mechanisms, is associated with a reduction in early and late TEPs [24,25]. Since PAS is believed to induce synaptic plasticity in a circuit that at least partially overlaps with SAI [26], we predicted that PAS would have induced TEP changes that are contrary to those induced by SAI and that persist beyond the duration of the stimulation paradigm. Furthermore, these changes in TEP components were expected to manifest already during the execution of the ongoing PAS protocol, offering new insight into the immediate impact of PAS-induced changes on cortical dynamics into the sensorimotor network.

To better characterize the relationship between PAS-induced corticospinal facilitation and cortical changes, we also investigated possible relationships between MEPs and TEPs measures.

## 2. Methods and Materials

### 2.1. Participants

Eighteen right-handed healthy subjects (27.3 ± 1.9 years; 8 females) participated in this study. All subjects were screened for any contraindications to TMS [27].

Participants provided written informed consent prior to participating in the study. All study procedures were approved by the institutional review board and were in accordance with the Declaration of Helsinki.

### 2.2. Electromyography (EMG)

EMG was recorded from the right first dorsal interosseous muscle (FDI), as target muscle, and the abductor pollicis brevis muscle (APB) muscle, as control muscle. EMG was recorded using bipolar Ag-AgCl surface electrodes (~2 cm apart) positioned in a belly-tendon montage. A ground electrode was fixed on the dorsum of the right hand. EMG was band-pass filtered at 10–1000 Hz, amplified 1000 times (Digitimer D360; Digitimer, Welwyn Garden City, UK), sampled at 5 KHz (CED 1401; Cambridge Electronic Design, Cambridge, UK), epoched around the stimulation pulse (−500 to 500 ms), and recorded on a computer for offline analyses.

### 2.3. Electroencephalography (EEG)

EEG was recorded from 64 scalp electrodes, positioned according to the international 10–20 system using a TMS-compatible amplifier (Bittium, NeurOne, Bittium Corporation, Finland) with a sampling rate of 2 KHz. The scalp electrodes site included Fp1, Fp2, F7, F3, Fz, F4, F8, FC5, FC1, FC2, FC6, T7, C3, Cz, C4, T8, FPz, CP5, CP1, CP2, CP6, PO9, PO5, P7, P3, Pz, P4, P8, FCz, O1, Oz, O2, AF7, AF3, AF4, AF8, F5, F1, F2, F6, TP9, FT7, FC3, FC4, FT8, TP10, C5, IZ, PO10, C6, TP7, CP3, CPz, CP4, TP8, P5, P1, P2, P6, PO7, PO3, POz, PO4, and PO8 and were mounted on the head with a cap (EASYCAP, Herrsching, Germany). The reference electrode was positioned on FCz, and the ground electrode was placed at the AFz. The electrodes were connected to the head using high-viscosity electrolyte gel (Abralyt HiCl, EASYCAP, Herrsching, Germany). All impedances were kept below 5 kΩ throughout the experimental sessions. During EEG recording, participants were asked to wear noise canceling headphones on top of earphones playing a noise specifically designed to mask the TMS click [28]. Raw EEG data were recorded and stored for offline analysis.

### 2.4. Electrical Nerve Stimulation

The interventional paired stimulation (interstimulus interval of 25 ms) was performed with ulnar nerve stimulation by using a constant current stimulator (Digitimer DS7AH, Digitimer Ltd., Hertfordshire, UK) with patients comfortably seated in an armchair. Superficial electrodes were placed on the skin over the ulnar nerve at the wrist. The cathode was positioned proximally, and the anode was positioned distally along the nerve’s course. Stimulation intensity was determined individually for each participant, starting from 2 mA and increasing in 0.5 mA steps. The minimal intensity required to evoke a reliable sensation for each participant was identified as the perceptual threshold.

### 2.5. TMS

A biphasic stimulator (Magstim SuperRapid^2^), connected to a real or sham figure-of-eight 70 mm diameter coil, was used to deliver TMS pulses. TMS was delivered over the left M1 on the scalp position eliciting the largest MEP in the contralateral FDI muscle. The coil was held tangential to the scalp at an angle able to induce a postero-anteriorly directed current perpendicular to the central sulcus. To constantly monitor the coil positioning over the hotspot, neuronavigation (Softaxic Optic, EMS, Bologna, Italy) with an optical tracking system (Polaris Vicra, Northern Digital, Waterloo, ON, Canada) was used in each participant. Sham stimulation was performed with a 70 mm figure-of-eight sham coil (Magstim Company Ltd., Whitland, UK) designed to produce an auditory percept similar to real TMS without cortical stimulation.

### 2.6. Experimental Paradigm

Experimental sessions were performed in the morning at the Department of Human Neuroscience, Sapienza University of Rome. Patients were seated on a chair designed for TMS (EMS, Italy), with their forearms resting on armrests. Subjects were instructed to keep their eyes open during the experiment. The optimal position of the coil was determined by a moderate suprathreshold stimulation intensity to constantly elicit the largest MEPs in the right resting FDI. At the optimal site, we determined the resting motor threshold (rMT) as the stimulator intensity needed to produce a minimal motor evoked response of at least 50 μV in the relaxed FDI in at least five of consecutive trials. We also defined the stimulator intensity sufficient to evoke a peak-to-peak MEP amplitude of 1 mV in the relaxed FDI (SI_1mV_).

PAS intervention was performed by pairing right ulnar nerve stimulation with TMS on left M1 with an interstimulus interval of 25 ms. The electrical nerve stimulation intensity during the PAS paradigm was set at 300% of the participant’s perceptual threshold, whereas TMS was delivered at SI_1mV_. The paired stimulation was repeated at a frequency of 0.1 Hz for a total of 180 pairs. We collected MEPs before (T0) and at 5 (T1), 15 (T2), and 30 (T3) min after PAS by delivering twenty TMS pulses at SI_1mV_ intensity, with an inter-pulse interval of 5 s with 15% jitter, during EMG recording. At T0 (pre-PAS) and T2 (post-PAS), we also collected real and sham TEPs by delivering, in two separate blocks, 100 real TMS pulses and 100 sham TMS pulses, with an inter-pulse interval randomly varied between 1.1 and 1.4 s [29], during EEG recording. Real TEPs were elicited at 110% rMT whereas sham stimulation intensity was set to match the subjective perception of real TMS. The order of real and sham TEPs blocks was pseudo-randomized across participants. The experimental paradigm included 15 min of resting between T0 measures collection and PAS intervention.

Finally, EEG was continuously recorded during PAS intervention to record TEPs elicited by the paired stimulation. To identify the sensory component related to peripheral stimulation during PAS intervention, EEG was also continuously recorded in a PAS control condition, involving only peripheral stimulation. In this control condition, participants received the same number and intensity of electrical stimulation as in the PAS condition. See, also, Figure 1.

### 2.7. Data Analysis

EMG data were processed offline using Signal Software V 6.0 (Cambridge Electronic Design, Cambridge, UK). The trials with pre-stimulus muscle contractions (up to 100 ms in the 500 ms preceding the TMS pulse) were identified and excluded from the analysis. We measured peak-to-peak MEP amplitude and computed the average across trials for each subject and time point.

TMS-EEG data (both pre-, post-PAS, and during PAS) were pre-processed using custom scripts in MATLAB (v 2017) using EEGLAB [30] and the TESA toolbox [31].

Preprocessing was in accordance with the established protocol by Rogasch et al. [31] and other previous studies [32,33,34,35].

Continuous EEG data were epoched from −1.4 s before to 1.4 s after the TMS, and epochs were demeaned considering the entire epoch length [31]. The TMS artifact was removed by cutting from 5 ms before to 10 ms after the TMS. The signal was then interpolated using cubic interpolation, and the data were resampled to 1000 Hz. Epochs contaminated by noise, movement, and EMG artifacts were removed by visual inspection. After re-referencing to average reference, a semiautomatic signal space projection for artifact removal (SSP-SIR) method was applied to suppress TMS-evoked muscle artifacts as implemented in TESA [36,37]. Epochs were then band-pass filtered from 1 to 100 Hz and band-stop filtered from 48 to 52 Hz using a 4th order Butterworth filter and shortened from −1 s before to 1 s after the TMS to avoid edges artifacts. We then run a round of independent component analysis (ICA) using the fastICA algorithm to remove artifact components related to residual TMS-evoked muscle, eye blinks and movements, muscle activity, and electrode noise. Lastly, data were transformed into reference-free current source density (CSD) estimations utilizing the “CSD” open-source Fieldtrip toolbox [38,39]. Final TEPs were obtained by averaging cleaned, CSD-converted, EEG epochs. To focus our analysis on left M1 local circuit dynamics, we averaged TEPs across a region of interest (ROI) including C1, C3, C5, CP3, and FC3 electrodes. We then computed the amplitude of P30, N45, P60, and N100 TEP components within this ROI by averaging the amplitude across the following time windows: 27–33 ms, 38–48 ms, 55–60 ms, and 100–131 ms. The time window for each component was defined based on the grand average TEPs computed across subjects and time points.

EEG data collected during peripheral nerve stimulation only and sham stimulation were pre-processed using the same methods and steps described for real TEPs but without applying the SSP-SIR and the need to remove TMS-evoked muscle artifacts with ICA. Final peripheral nerve- and sham-evoked potentials were obtained by averaging cleaned, CSD-converted, EEG epochs.

The SSP-SIR function was further applied on final TEPs recorded during PAS to suppress the sensory component related to peripheral stimulation using the peripheral nerve-evoked potentials as control signal. For this purpose, final peripheral nerve-evoked potentials were time-shifted by 25 ms to match the TEPs time course.

### 2.8. Statistical Analysis

Statistical analysis was performed by using IBM SPSS Statistics for Windows, Version 25.0. Armonk, NY, USA: IBM Corp. 25.

To investigate PAS-induced effects on MEPs, a two-way repeated measure ANOVA was conducted to analyze the differences in MEP amplitude between the two muscles (FDI, APB) and across the four time points (Pre-PAS, T1 Post-PAS, T2 Post-PAS, T3 Post-PAS). To investigate PAS-induced effects on TEPs, a three-way repeated measures ANOVA was conducted to examine the effects of Condition (Real, Sham), Component (P30, N45, P60, N100), and Time (Pre-PAS, Post-PAS). To examine the modulation of TEPs component during the PAS paradigm, the PAS session was divided into four equal quartiles, containing the same number of TMS pulses (PAS I, PAS II, PAS III, PAS IV). A two-way repeated measures ANOVA was conducted to examine the effect of Component (P30, N45, P60, N100) and Time (PAS I, PAS II, PAS III, PAS IV) as well as the interaction between Component and Time.

Spearman’s rank correlation coefficient was applied to assess potential correlations between PAS-induced effects on MEPs and TEPs computed as the difference (Delta) between the mean amplitude at different time points after PAS to the mean amplitude before. Furthermore, a linear regression analysis was conducted to explore the ability of PAS-induced effects on TEPs to predict MEP facilitation. A value of *p* < 0.05 denoted statistical significance. To correct for multiple comparisons, when necessary, the False Discovery Rate (FDR) correction was employed. Data were expressed as mean ± standard deviation unless otherwise specified.

## 3. Results

All eighteen healthy volunteers completed the study procedures without any adverse events reported.

### 3.1. Effect of PAS on MEPs

The mean MEP amplitudes at each time point for FDI and APB muscle are reported in Table 1.

ANOVA on MEPs showed a significant main effect of Muscle (F = 20.95, DFn = 1, DFd = 32; *p* < 0.0001) and Time (F = 19.26, DFn = 3, DFd = 96; *p* < 0.0001) and a significant interaction between Muscle and Time (F = 17.10, DFn = 3, DFd = 96; *p* < 0.0001).

In the FDI muscle, MEP amplitudes significantly increased after PAS session at T1 (mean difference = 0.6154; *p* = 0.0008), T2 (mean difference = 0.8650; *p* < 0.0001), and T3 (mean difference = 0.6558; *p* < 0.0001). In contrast, no significant changes in MEP amplitude were observed in the APB muscle at any time point compared to baseline (all *p* > 0.05).

### 3.2. Effect of PAS on TEPs

There was a significant main effect of Condition (F(1,15) = 12.035, *p* = 0.003, η^2^_partial = 0.445), Component (F(3,45) = 13.547, *p* < 0.001, η^2^_partial = 0.475), and Time (F(1,15) = 4.679, *p* = 0.047, η^2^_partial = 0.238), as well as a significant interaction between Condition and Component (F(3,45) = 11.637, *p* < 0.001, η^2^_partial = 0.437), Condition and Time (F(1,15) = 8.469, *p* = 0.011, η^2^_partial = 0.361), and Component and Time (F(3,45) = 4.791, *p* = 0.006, η^2^_partial = 0.242). Importantly, the three-way interaction between Condition, Component, and Time was significant (F(3,45) = 6.493, *p* = 0.001, η^2^_partial = 0.302).

For the Real condition, we found significant effects of Component (F(3, 48) = 13.526, *p* < 0.001, partial η^2^ = 0.458), Time (F(1, 16) = 6.585, *p* = 0.021, partial η^2^ = 0.292), and the interaction between Component and Time (F(3, 48) = 4.888, *p* = 0.005, partial η^2^ = 0.234). Post-hoc revealed significant increases in the P30 (pre-PAS: 7.33 ± 4.09; post-PAS: 21.80 ± 4.84; *t*: −2.96; *p* = 0.009) and P60 (pre-PAS: 13.51 ± 6.36; post-PAS: 24.27 ± 8.16; *t*: −2.22; *p* = 0.041) component following the PAS intervention. The N45 (pre-PAS: 6.37 ± 3.83; post-PAS: 16.76 ± 3.78; *t*: −2.01; *p* = 0.061) component displayed an increase, and the N100 component showed a decrease (pre-PAS: −16.45 ± 4.94; post-PAS: −21.73 ± 3.94; *t*: 1.85; *p* = 0.083) but these changes were not significant (Figure 2).

For the Sham condition, a significant within-subjects effect was found only for Time (F(1, 15) = 5.156, *p* = 0.038, partial η^2^ = 0.256) with a significant decrease in amplitude at post-PAS compared to pre-PAS (mean difference = 1.129, *p* = 0.038) (Figure 3).

### 3.3. Modulation of TEPs during the PAS Paradigm

The analysis revealed a significant main effect of Component (F(1,16) = 7.020, *p* = 0.004, η^2^_partial = 0.601) as well as a significant interaction between Component and Time (F(1,16) =5.476, *p* = 0.013, η^2^_partial = 0.860). There was no significant main effect of Time (F(1,16) = 0.703, *p* = 0.566, η^2^_partial = 0.131). Follow-up analysis showed no significant effect of Time on any Component (all *p* > 0.05) (Figure 2).

### 3.4. Relationship between PAS-Induced Effects on MEPs and TEPs

Correlation analysis was performed only considering MEP changes at T2 (i.e., maximal effect) and TEPs measures significantly influenced by PAS intervention. Our results revealed no significant correlation between PAS-induced effects on MEPs and TEPs (all *p* > 0.05).

A linear regression analysis was conducted to examine the influence of Delta_P30 and Delta_P60 on Delta_MEP_T2. The linear regression analysis revealed that the model based on changes in TEP P30 and P60 significantly explained variability in PAS-induced changes in MEPs at T2 (F(2,14) = 4.329, *p* = 0.034), accounting for 38.2% of the variance (adjusted R^2 = 0.294, standard error of the estimate was 0.40328). The individual predictors, Delta_P30 (B = 0.009, SE = 0.006, β = 0.358, *t* = 1.517, *p* = 0.152) and Delta_P60 (B = 0.009, SE = 0.006, β = 0.367, *t* = 1.557, *p* = 0.142), were not statistically significant at the 0.05 level.

## 4. Discussion

In this study, we examined the cortical correlates of a standard PAS 25 paradigm by evaluating the impact of the intervention on both TEP components and MEP amplitudes in a cohort of eighteen healthy subjects. We found that PAS intervention induced the expected long-lasting facilitatory modulation of MEP amplitude. In addition, PAS induced a significant increase in TEPs P30 and P60 amplitude from M1 stimulation while no significant modulation was observed for N45 and N100 TEP components. We did not find any significant correlation between the magnitude of PAS-induced changes in TEP components and MEP amplitude. However, PAS-induced combined changes of P30 and P60 amplitude accounted for 38% of the explained variability in PAS aftereffects on MEP amplitude.

We took proper methodological precautions to control for potential confounding factors that could have influenced our study results. PAS intervention was performed according to previous studies [40,41]_._ During the experimental session, a neuronavigation system was used to exclude possible bias due to unstable coil positioning. The use of noise masking during TEPs recording and the inclusion of a sham condition limits the possibility of confounding due to TMS-evoked auditory evoked potentials. We scheduled the experimental session in the morning to avoid any potential impact of fatigue on the day of testing. Finally, all participants underwent a thorough screening for neurological and psychiatric disorder by a trained neurologist, and those taking medications known to affect PAS effects (corticosteroids, anxiolytics, centrally acting ion channel blockers, or antihistamines) were excluded from the study [6].

Our study confirms a significant increase in FDI MEP amplitude following a PAS-25 protocol [1,3,40]. PAS has been proposed to induce a form of Hebbian-like synaptic plasticity in the human motor cortex [2,3]. Our study found that several properties characterizing the classic model of associative Hebbian learning were present in the PAS-induced effects we observed. Specifically, the facilitation of MEP amplitude induced by PAS exhibited a rapid development, becoming evident as early as 5 min post-intervention and persisting for up to 30 min after the intervention ceased. Furthermore, our findings support the notion that PAS has a topographically specific effect on corticospinal excitability, with the effect being observed in the FDI muscle, which is an intrinsic muscle of the hand innervated by the ulnar nerve, and not in the APB muscle, which is a muscle innervated by the median nerve. Overall, our findings support the validity of the PAS paradigm as a neurophysiological tool to induce a rapid and enduring change in excitability in the corticospinal output circuitry, as measured by MEPs amplitude.

The novelty of our study lies in the methodological approach we employed to directly measure the plasticity effect induced by PAS at the cortical level. Specifically, we assessed PAS-induced changes in M1 local dynamics by measuring TEPs as an outcome measure. Our study revealed that PAS induced differential effects on TEP components, with a significant facilitation of the P30 and P60 amplitude and no significant changes of N45 and N100. We also recorded TEPs evoked by the PAS protocol to gather insights into the immediate effects of the intervention on M1 excitability and the time course of any cortical plasticity changes. Despite the lack of significant post hoc effects, we observed a trend for a progressive increase in P30 and, in particular, the P60 component throughout the PAS paradigm. We exclude the possibility that a lack of significant facilitation of TEPs during PAS may be due to a ceiling effect caused by the high stimulation intensity since the amplitude of TEPs measured after PAS was even higher.

The facilitation of P30 after PAS is consistent with previous observations suggesting that the P30 reflects local circuits excitatory neurotransmission [8,9,18,42,43,44]. Our study further demonstrated a significant facilitation of the P60 component following the PAS intervention, with a noticeable trend for a progressive increase in its amplitude observed during the PAS paradigm. The P60 generator is slightly more posteriorly shifted toward the postcentral gyrus than the P30 [8,45,46], but P60 amplitude showed to be similarly affected by TMS protocols known to modulate M1 excitability [16,44]. It is important to note that the P60 component may partly reflect afferent proprioceptive signals associated with the MEP [8,47,48]. Thus, the facilitation of P60 following PAS may be partially attributed to a larger MEP-associated afferent volley. However, P60 changes has been reported also with subthreshold intensity, suggesting other possible mechanisms underlying the PAS-induced modulation of its amplitude [48]. Further investigation with better control for reafferent inputs is needed to clarify the cortical origin of the PAS-induced P60 facilitation.

We found a lack of correlation between the magnitude of PAS-induced facilitation of MEP amplitude and TEPs peaks amplitude. The relationship between corticospinal measures such as MEPs and proper cortical measures such as TEPs has long been a subject of debate [49,50]. The lack of correlation may reflect the different neural mechanisms underlying TEPs and MEPs [51,52]. While TEPs reflect summation of postsynaptic potentials over a large population of cortical neurons, MEPs arise from the activation of a smaller population of neurons directly projecting on the pyramidal tract neurons. Furthermore, unlike TEPs, MEPs are also significantly influenced by spinal cord mechanisms. However, despite the lack of correlation, both measures were facilitated post-PAS, and the linear regression analysis revealed that the combined changes in P30 and P60 component amplitudes significantly predicted the MEP facilitation after PAS. Taken together, correlation and regression results suggest a complex relationship between TEPs and MEPs changes after PAS. Changes in TEP amplitudes may reflect a cortical mechanism that indirectly contributes to corticospinal facilitation, or TEPs and MEPs changes may simply share a common cause being independent aftereffects of PAS-induced plasticity on distinctive neural populations.

We acknowledged several limitations. A larger sample size would have provided greater statistical power to detect significant differences and correlations, increasing the generalizability of our results. Another limitation is the employment of different stimulation intensities for recording TEPs as opposed to MEPs. This discrepancy in stimulation intensities between MEPs and TEPs may have introduced variability in the measured responses, potentially affecting the strength of the observed correlations.

Finally, interpreting the meaning of specific TEP components should be approached with caution due to an incomplete understanding of their neurobiological basis and generators.

In conclusion, our study provides novel insight into the neurophysiological changes associated with PAS-induced plasticity at M1 cortical level. Furthermore, our findings support the validity of the PAS paradigm as a robust neurophysiological tool to induce corticospinal plasticity. Further research involving larger cohorts and extending to disease models where PAS-induced plasticity is altered, such as dystonia or Parkinson’s disease, are warranted for a deeper understanding of the underlying mechanisms and potential clinical implications of these modulations.

## Figures and Tables

**Figure 1 brainsci-13-00921-f001:**
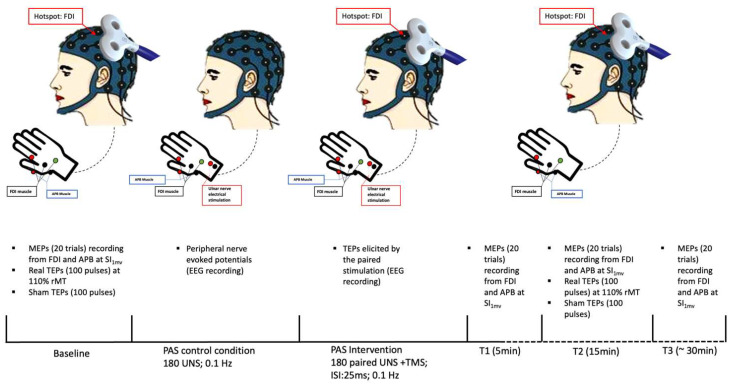
Experimental paradigm. FDI: first dorsal interosseus muscle; APB: abductor pollicis brevis muscle; MEPs: motor-evoked potentials; SI_1mV_: stimulator intensity sufficient to evoke a peak-to-peak MEP amplitude of 1 mV; TEPs: transcranial magnetic stimulation-evoked potentials; rMT: resting motor threshold; UNS: ulnar electrical nerve stimulation; ISI: interstimulus interval.

**Figure 2 brainsci-13-00921-f002:**
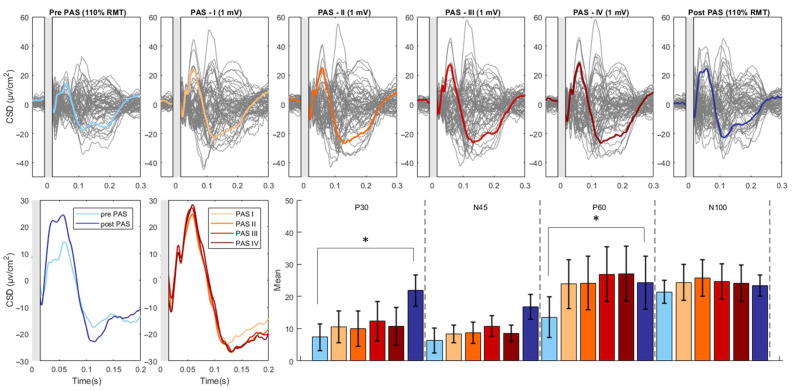
PAS-related modulation of TMS-evoked potentials. The upper part of the figure illustrates the butterfly plots of the real TEPs obtained by averaging cleaned, CSD-converted, EEG epochs for each time point of the study (average of all 18 participants). The lower left part of the figure displays the differences in CSD estimates before and after the PAS intervention as well as during the PAS session (PAS session was divided into four equal quartiles, containing an identical number of TMS pulses). The lower right part of the figure reports the mean amplitude of each TEP component throughout the various time points of the study. The asterisk denotes a statistically significant difference (*p* < 0.05). (CSD: current source density).

**Figure 3 brainsci-13-00921-f003:**
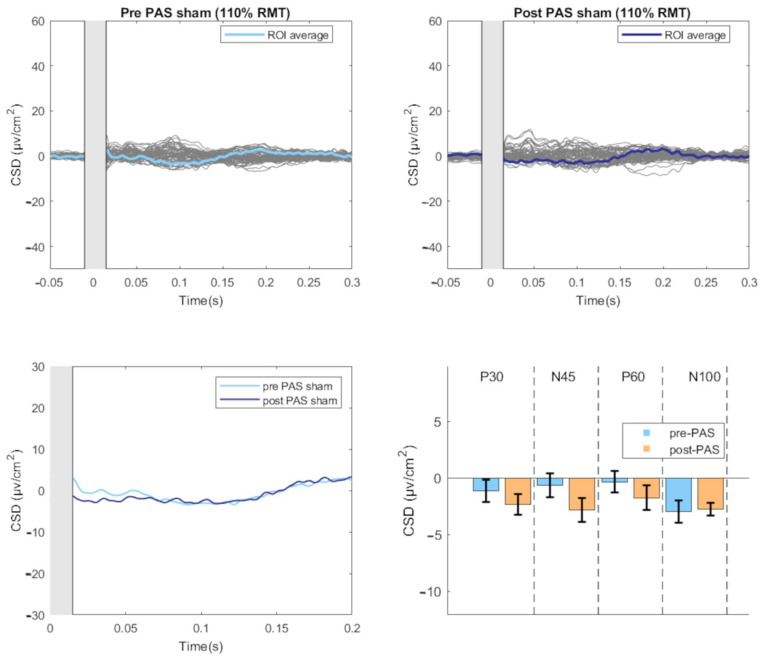
TEPs-evoked by sham stimulation before and after the PAS intervention. The upper part of the figure illustrates the butterfly plots of the sham TEPs, obtained by averaging cleaned, CSD-converted, EEG epochs, before and after PAS session (average of all 18 participants). The lower left part of the figure displays the differences in CSD estimates before and after the PAS intervention. The lower right part of the figure reports the mean amplitude of each TEP component before and after PAS intervention. (CSD: current source density).

**Table 1 brainsci-13-00921-t001:** Mean MEP amplitudes and standard deviations for first dorsal interosseus (FDI) and abductor pollicis brevis (APB) muscle at different time points.

Time Course	FDI Muscle		APB Muscle	
	Mean Amplitude (mV)	Std. Deviation	Mean Amplitude (mV)	Std. Deviation
Baseline	1.229	0.393	0.881	0.432
T1 Post-Pas	1.844	0.862	0.917	0.532
T2 Post-PAS	2.094	0.724	0.909	0.476
T3 Post-PAS	1.885	0.707	0.882	0.450

## Data Availability

The data presented in this study are available upon reasonable request.

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
