# Peer review of "Motor Cortical Correlates of Paired Associative Stimulation Induced Plasticity: A TMS-EEG Study"

_brainsci, 2023, doi:10.3390/brainsci13060921_

Round 1
Reviewer 1 Report
Review for manuscript brainsci-2411435, "Motor cortical correlates of paired associative stimulation induced plasticity: a TMS-EEG study" by Costanzo et al.
This study is scientifically well-performed and very well-written. The methods are performed well and the results comprehensible. The discussion addresses the important issues and the conclusions cover the research question and the main findings.
I have no issues and recommend publication in Brain Sciences.
Author Response
We thank the reviewer for the positive feedback and recommendation for publication. We truly appreciate its time and effort in reviewing our study.
Reviewer 2 Report
Dear authors,
I have found the projects interesting due to the clarity of the methodological approach in an attempt to demonstrate PAS-induced plasticity at the M1 cortical level in healthy subjects vs control.
The abstract reveals concise, informative data regarding the aim, type, and study results.
• The material and methods succinctly present the settings of the machines (EEG, EMG and TMS) and procedures protocol followed in this study. In addition, the experimental paradigm describes the protocol followed for the active subjects and controls.
• Data analysis explains the phases of pre and post-processing of the collected data for each study analysis is described separately.
• The results are adequately presented and analysed in a stratified approach, as the authors discussed the effect of PAS on both MEP and TEP in real and sham conditions.
• The results were discussed according to the study's objectives and the existing theories about Paired Associative Stimulation PAS effect on motor cortex excitability, plasticity, and learning.
• The authors
• reviewed the measures considered to control the potential confounding factors related to the acquisition and processing of the data
• the limitation of data
I believe the article is a valuable tool for assessing the strengths and weaknesses of a larger project, as it is well-structured and logical and should be published in its present form.
Author Response
We sincerely appreciate reviewer’s insightful feedback and supportive comments. The detailed recognition of the study's methodological approach, analysis, and discussions is greatly valued.
Reviewer 3 Report
The presented study is interesting and significant.
However, the introduction is very short and insufficient in order to present the background, novelty and aims of the study. Combination of TMS and EEG and TEPs are novel approaches conducted by limited number of research groups and in a limited number of studies. Since the methods are novel and not thoroughly explored, authors should give a review on previous relevant studies and their results and provide reasoning why using TES for assessment of the effects and provide clear hypotheses based on previous findings. Number of references regarding TMS-EEG and TES recording and processing needs to be significantly broadened.
Next methodological issue is the data processing of TESs.
Authors mention epochs of -1.4 to 1.4 s around the stimulus, baseline correction on the whole epoch and other processing steps which are highly unusual for standard EPs and ERPs. Authors do that without providing any reference or reasoning for this approach. Baseline removal using the whole epoch can significantly affect the post stimulus results and create false statistical differences. Standard method for baseline removal is the average value of pre-stimulus interval.
Generally, each decision regarding every processing step should be either backed up with references to previous studies or thoroughly elaborated since TES as a novel method still doesn't have clear processing recommendations so new studies should either try to replicate previous methods for comparison or if introducing new methods, clearly mention this and provide the reasoning for the choice.
Round 2
Reviewer 3 Report
Authors have addressed my comments adequatly